# Biological and Physical Effects of Brine Discharge from the Carlsbad Desalination Plant and Implications for Future Desalination Plant Constructions

**Karen Lykkebo Petersen** [1,2,*], **Nadine Heck** [3,4], **Borja G. Reguero** [3,4], **Donald Potts** [3,5], **Armen Hovagimian** [6] and **Adina Paytan** [3]

1   Department of Earth and Planetary Sciences, University of California, Santa Cruz, 1156 High Street, Santa Cruz, CA 95064, USA

2   Department of Ecology, Environment and Plant Science, Stockholm University, Svante Arrhenius väg 20A, 114 18 Stockholm, Sweden

3   Institute of Marine Sciences, University of California, Santa Cruz, 1156 High St., Santa Cruz, CA 95064, USA; nheck@ucsc.edu (N.H.); borja_reguero@tnc.org (B.G.R.); potts@ucsc.edu (D.P.); apaytan@ucsc.edu (A.P.)

4   The Nature Conservancy, 115 McAllister Way, Santa Cruz, CA 95060, USA

5   Department of Ecology and Evolutionary Biology, University of California, Santa Cruz, 130 McAllister Way, Santa Cruz, CA 95060, USA

6   Cowell College, University of California, Santa Cruz, 1156 High Street, Santa Cruz, CA 95064, USA; ahovagim@ucsc.edu

*   Correspondence: lykkebo.petersen@su.se

**Abstract:** Seawater reverse osmosis (SWRO) desalination is increasingly used as a technology for addressing shortages of freshwater supply and desalination plants are in operation or being planned world-wide and specifically in California, USA. However, the effects of continuous discharge of high-salinity brine into coastal environments are ill-constrained and in California are an issue of public debate. We collected in situ measurements of water chemistry and biological indicators in coastal waters (up to ~2 km from shore) before and after the newly constructed Carlsbad Desalination Plant (Carlsbad, CA, USA) began operations. A bottom water salinity anomaly indicates that the spatial footprint of the brine discharge plume extended about 600 m offshore with salinity up to 2.7 units above ambient (33.2). This exceeds the maximum salinity permitted for this location based on the California Ocean Plan (2015 Amendment to Water Quality Control Plan). However, no significant changes in the assessed biological indicators (benthic macrofauna, BOPA-index, brittle-star survival and growth) were observed at the discharge site. A model of mean ocean wave potential was used as an indicator of coastal mixing at Carlsbad Beach and at other locations in southern and central CA where desalination facilities are proposed. Our results indicated that to minimize environmental impacts discharge should target waters where a long history of anthropogenic activity has already compromised the natural setting. To ensure adequate mixing of the discharge brine desalination plants should be constructed at high-energy sites with sandy substrates, and discharge through diffusor systems.

**Keywords:** SWRO desalination; brine discharge; osmotic stress; coastal monitoring; impacted coastal systems

## 1. Introduction

Freshwater demand is increasing worldwide due to a variety of factors, including population growth, agricultural expansion and environmental changes [1]. At the same time, natural freshwater resources are declining in quantity and quality [2]. In California, increasing agricultural activity and population growth have diminished natural groundwater reservoirs, resulting in substantial land subsidence and seawater intrusion [3]. Recurrent droughts that limit recharge of water reservoirs are common and are expected to increase in frequency, duration and intensity [4,5] exacerbating the problem. Seawater desalination by reverse osmosis (SWRO) is increasingly being seen as a way to counter freshwater shortages, and several desalination plants are being built or proposed along the Southern and Central California coastline [6,7]. Ten small seawater reverse osmosis (SWRO) desalination facilities with individual capacities from $30 \times 10^3$ to $10 \times 10^6$ L day$^{-1}$ (combined capacity ~$20 \times 10^6$ L day$^{-1}$) [8] are currently operating along the California coast, and one large-scale plant with a capacity of $180 \times 10^6$ L day$^{-1}$, started operation in Carlsbad (Southern California) in December 2015. Seven additional large-scale facilities have been proposed (one is under construction) with capacities of $40–500 \times 10^6$ L day$^{-1}$ [8] in response to reoccurring droughts and increasing demand for water resources.

SWRO facilities draw coastal water as feed and continuously produce high salinity brine effluent that is typically diluted and discharged back into coastal environments. The discharge occurs either directly at the shoreline through outfall channels, or further from shore through pipes or diffusor systems [9–11]. Californian regulations require that the brine discharged not exceed two salinity units above ambient levels within 100 m offshore from the discharge point [12]. The efficiency of water mixing and the footprint of the discharged brine depend on (1) dilution prior to discharge and hence the final density of the discharging brine, (2) local coastal conditions (waves, currents and bathymetry), and (3) the design of the discharge method (channel, pipe or diffusor). To comply with these regulations, SWRO facilities need to control the salinity of the brine (by dilution prior to discharge) and select a discharge design that ensures easy mixing of brine with the surrounding seawater under the specific oceanographic conditions at the discharge location.

Water mixing potential at the discharge zone is usually evaluated prior to operation by using hydrodynamic computer models to ensure compliance [12]. While the California Ocean Plan (2015 Amendment to Water Quality Control Plan) specifies a salinity impact zone extending no more than 100 m offshore, the Carlsbad Desalination Plant received an exception to policy to extend its salinity impact zone to 200 m offshore, due to its high capacity [12]. For the Carlsbad Plant, the hydrodynamic model assumed a starting salinity of 42 at the outfall, and predicted that salinity would decline to 35.5 (ambient 33.2) at a distance of 196 m offshore (i.e., ~2 salinity units above ambient within the 200 m permitted limit) [13]. The brine produced at the Carlsbad SWRO facility is diluted by mixing with seawater used for cooling at a co-located power plant, and hence decreasing the brine salinity and increasing water temperature, hence reducing the brine density to increase the mixing potential and prevent bottom ponding of a high density brine.

Despite increasing use of SWRO desalination worldwide as well as in California, impacts of brine effluent discharge on the living organisms and ecosystems in the coastal environments are ill-constrained [14,15]. Past research on pelagic phytoplankton and benthic microbes, seagrasses, polychaetes and corals demonstrate that salinity tolerances are highly variable among species and also dependent on the magnitude of the salinity increase and exposure time [14,16–21]. For instance, seagrasses have low thresholds with a detectable mortality at salinity of 5% above ambient levels, whereas coral growth is not impacted at salinity as high as 10% above ambient [22–24]. Relative abundances and growth rates of phytoplankton, zooplankton, and benthic bacteria also do not seem to be significantly impacted at salinities of 10% above ambient, but community structure often changes [16,17,25,26]. The Benthic Opportunistic Polychaetes and Amphipods index (BOPA-index) is commonly used as an indicator of the level of "disturbance" to benthic communities in areas impacted by pollution [27–30], or other stressors such as changes in salinity [31]. This index is

based on an inverse relationship between the abundances of sensitive amphipods and opportunistic polychaetes [27,29,30,32]. The index value specifies an ecological status ranging from "good" to "bad", where good is defined as an area dominated by sensitive species, and bad an area dominated mainly by opportunistic species.

Coastal California is a highly productive zone supported by upwelling of nutrient-rich sub-surface water. This productivity supports large kelp beds, productive rocky reefs with high biodiversity, and rich plankton communities that serve as food for numerous fish, seabirds, whales and dolphins. However, the population density along the coast in California is high (26 million people living in coastal counties) and many costal settings have been impacted by a wide range of anthropogenic activities including dredging, shipping, sewage discharge, eutrophication, commercial and recreational fishing and more transitional waters [33]. Along the state coastline, 124 marine protected areas (MPA's) are providing refuge for the ecosystem (total coverage of 16% of state water) [34,35]. Toxicity testing with high-salinity seawater has been conducted on a few key Californian rocky-reef species (i.e., *Haliotis rufescens* (Red Abalone), *Strongylocentrotus purpuratus* (Purple Urchin) and *Dendraster excentricus* (Sand Dollar)) and all proposed SWRO desalination facilities are required to use hydrological modeling to estimate the impact area of discharging brine [12,36]. However, uncertainties and concerns persist regarding potential impacts of SWRO desalination brine on coastal environments, especially among coastal users and the general public [37,38].

This study characterizes the spatial footprint of the discharge plume from Carlsbad Desalination Plant, both chemically and biologically, with the ultimate goal of informing legislators, regulators, plant managers and the public about appropriate locations and discharge methods for future SWRO plants and illustrates the need for monitoring. The extend and impact of the discharge zone is characterized by using (1) water samples collected at and around the outfall channel of the Carlsbad Desalination Plant, before and after the plant became operational (in December 2015), (2) biological surveys of benthic epifauna, (3) a BOPA analysis around the discharge zone, and (4) a laboratory bioassay with brittle stars (*Ophiothrix spiculata*). The study finds the brine plume to extend beyond the 200 m impact zone allowed in the California Ocean Plan (2015 Amendment to Water Quality Control Plan) but finds no significant impact on the benthic ecology. Using a model of coastal wave energy at Carlsbad Beach and in Southern and Central California, possible impact of future desalination plants in these California coastal zones is assessed.

## 2. Methods

### 2.1. Study Area

The Carlsbad Desalination Plant, built and operated by Poseidon Water, is the first and currently only large-scale SWRO desalination facility in California. Operations began in December 2015 with a capacity of $180 \times 10^6$ L day$^{-1}$. The plant is located in an industrial area at the southern end of the Agua Hedionda Lagoon, adjacent to the Encina Power Station (Figure 1A). Seawater enters the lagoon through a dredged channel at the north end of the lagoon, about 1 km from the seawater intake used by the power station as cooling water (since 1954). The desalination plant diverts ~10% of this water released by the power plant post-cooling for SWRO (Figure 1B). The brine effluent is then returned to the cooling water further downstream, resulting in a 1:10 dilution of the brine before the mixture is discharged to the ocean through a 10–15 m wide channel (outfall) between two rocky walls extending ~50 m offshore at Carlsbad Beach (Figure 1A) [39]. Carlsbad Beach is a relatively high-energy beach with wave heights averaging ~2.5 m in winter and ~1.5 m in summer [40–43]. The nearshore habitat is dominated by a sandy bottom with scattered small rocky reefs and seagrass patches at the northern end of the beach. The nearshore region is shallow with depths of 5–10 m up to 800 m offshore, and with deeper water (~20 m) starting ~1 km offshore [41].

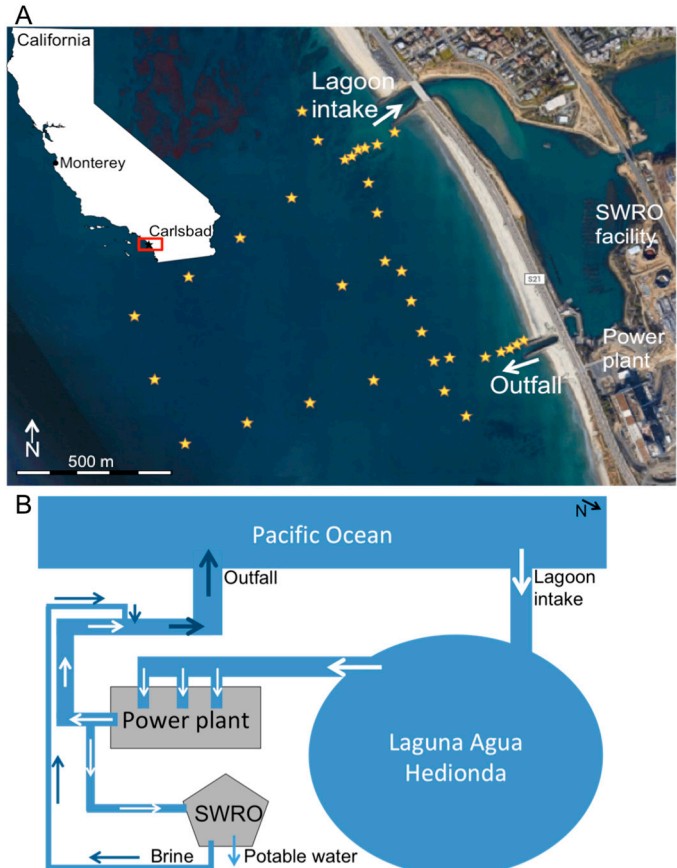

**Figure 1.** (**A**) Aerial view of the study site around Carlsbad Desalination Plant. The SWRO plant and power plant, intake and outfall channels, and sampling sites (stars) are marked; (**B**) schematic of water flow from the Pacific Ocean to Laguna Aqua Hedionda and the intake and discharge of water by the power plant and SWRO facility (not to scale). White arrows indicate seawater flow and dark-blue arrows represent discharging brine and its mixture with the power plant's seawater return flow.

## 2.2. Sample Collection and Biological Surveys

Water and sediment samples were collected, and benthic surveys were conducted twice in the year prior to the start of SWRO operation (pre-operation), and twice in the first year after brine discharge began (post-operation). Pre-operation samples were collected in December 2014 and September 2015, and post-operation samples were collected in May (five months post-operation) and November 2016 (11 months post-operation). Samples were collected by scuba divers, starting ~100 m offshore (closer if conditions allowed). Samples were taken along two transects perpendicular to the beach at 25 m intervals out to 200 m, and then at 250, 300, 400, 600, 800 and 1000 m from shore. Sampling was conducted close to shore as the chemical footprint and possible biological impacts were expected to be greatest close to the discharge site and quickly dissipate with distance due to mixing. A transect parallel to shore was conducted as close to the beach as possible (~200 m offshore), with samples collected every 100 m from 200 m south of the outfall to 200 m north of the lagoon entrance (Figure 1A). Bottom and surface water samples were collected at each sampling site in 1 L acid-cleaned, sample-rinsed, Nalgene HDPE bottles. Water temperatures were measured on the boat, immediately after collection, with a YSI 30SCT probe (YSI, Yellow Springs, OH, USA). Water samples were kept in a cooler until they were filtered (using various filters, see Section 2.3 below) and processed, within 4 h after collection. Sediment samples (from the upper 10 cm) were collected in 250 mL plastic jars by pushing the jars into the sediment and samples were kept frozen until analysis (see Section 2.3).

Benthic surveys of epifauna were conducted along two continuous perpendicular transects each 1 m wide, from ~100 to 200 m offshore, and then extended with $1 \times 1$ m quadrats (1 m$^2$) at ~100 m intervals from 200 to 1000 m offshore. Along the continuous transects, a 100 m measuring tape was laid on the sea floor and all visible benthic organisms within 0.5 m of each side of the tape were recorded for each 10 m long section (10 m$^2$). At stations running parallel to shore, organisms were recorded in ten randomly distributed 1 m$^2$ quadrats per station. Visible benthic epifauna were enumerated separately by two scuba divers, and counts were calculated to species abundance per m$^2$ and grouped in counts close to the discharge channel (0–200 m offshore) and in the surrounding area (250–1000 m offshore). Organisms were assigned to 10 categories: Porifera (sponges), Anthozoa, Gastropoda, Bivalvia, Cephalopoda, Polychaeta, Echinodermata, Arthropoda, fish, macroalgae and seagrass.

*2.3. Water and Sediment Analyses*

Water samples were analyzed for salinity, chlorophyll *a*, nutrients (PO$_4$, NO$_3$ and SiO$_2$), particulate organic matter (POM), dissolved organic carbon (DOC) and trace metals. More detail is provided below:

Salinity: 50 mL of seawater were pumped through a Guildline Portasal salinometer (Osil, Hampshire, UK) and salinity recorded with an accuracy of $\pm 0.001$.

Chlorophyll *a*: 250 mL of seawater were filtered onto GF/F filters (0.7 μm) (Whatman, Chicago, IL, USA) and kept frozen in the dark until further analysis. The filters were immersed in vials with 7 mL of 90% acetone and extracted at 2 °C for 12 h before measurements. Chlorophyll *a* (Chl *a*) concentrations were measured using a TD-700 fluorometer (Turner Designs, San Jose, CA, USA) calibrated with Chl *a* according to the EPA 445 method, using CS-5-60 and CS-2-64 glass filters (Turner Designs, San Jose, CA, USA).

Nutrients: 50 mL of seawater were filtered through 0.2 μM filters and kept frozen until analysis. Samples were analyzed for soluble reactive PO$_4$ (SRP) [44] and for NO$_3$ and SiO$_2$. Samples were processed on a QuikChem FIA+ 8000 Series autoanalyzer (Lachat Instruments, Milwaukee, WI, USA) using methods 31-115-01-3-A (PO$_4$), 31-114-27-1-B (NO$_3$) and 31-107-04-1-A (SiO$_2$). Sample reproducibility was 0.5%, with a detection limit of 0.15 μM for all analytes.

Dissolved organic carbon: 30 mL of seawater were collected in pre-ashed glass vials (450 °C for 4 h). The seawater was acidified to pH 2 using concentrated trace metal grade HCl, and analyzed on a Shimadzu TOC-V autosampler (Shimadzu, Columbia, MD, USA) with prepared standards of 2.125 g KHP L$^{-1}$. Detection limit was 0.3 μM with 2 mL injections.

Particulate organic matter: 400 mL of seawater were filtered onto pre-ashed (450 °C for four hours) GF/F filters (0.7 μm) and samples kept frozen. The filters were dried for 48 h at 50 °C and pressed into tin capsules. The carbon to nitrogen molar ratio (C:N) and C and N isotope ratios were measured using isotope ratio mass spectrometry (CE instruments NC2500, Wigan, UK). Analyses were done with standards of pugel and acetanilide to an accuracy of $\pm 0.11$‰.

Select major and trace elements: 50 mL of seawater was filtered (0.2 μM) and collected in acid-cleaned sample-rinsed LDPE bottles and acidified with trace metal grade HCl to pH 2 prior to analysis. Samples were diluted 500-fold and analyzed on an Element XR ICP-MS (Thermo Fisher Waltham, MA, USA) with standard curves prepared using NIST 1640a standard solution (National Institute of Standards and Technology, Gaithersburg, MD, USA). The average of two procedural blank values was used for blank correction. Concentrations of Li, Na, Mg, K, Ca, Mn, Fe, Sr and Ba were measured.

Infaunal organisms: Sediment samples were thawed and washed with 90% ethanol. The ethanol rinse was collected and all polychaetes and amphipods were counted immediately under a dissecting microscope. For polychaetes, individuals of the two most abundant families, Paraonidae and Capitellidae, were identified and counted; individuals from other families were counted as "other". For amphipods, the families Gammaridae, Hyperiidae, Caprellidae and Corophiidae were identified and counted. A BOPA-index was calculated for each sediment sample using methods described

in reference [27] (Equation (1)), where $fp_{op}$ is the proportion of polychaetes and $f_a$ is the proportion of amphipods:

$$BOPA = log\left(\frac{fp_{op}}{f_a + 1} + 1\right). \tag{1}$$

Sediment grain size: The sediment was dried, and 75–90 g subsamples were sorted using twelve sieves (from 0.5 Φ to 4.75 Φ, Krumbein phi scale), and each fraction was weighed. The mean grain size and sorting factor were determined using methods described in [45].

### 2.4. Bioassay Experiment

A bioassay was conducted with brittle stars (*Ophiothrix spiculata*) collected in Monterey Bay (collected by Monterey Bay Abalone Co) and exposed to discharged desalination brine collected from the outfall channel of the Carlsbad Desalination Plant. The collected brine was filtered through a 0.2 μM filter and kept cool and in the dark with aquarium air pumps attached to maintain oxygen levels. Brittle stars were exposed to three different brine treatments: (1) Discharge water of salinity 37 collected from the discharge channel; (2) 1:1 dilution of the discharge water with ambient seawater to a salinity of 34; and (3) ambient seawater (control) with salinity of 33. There were 20 replicates for each treatment. Each replicate consisted of two, randomly selected, brittle stars maintained in a 500 mL glass culture dish containing 450 mL of treatment water and capped with a glass Petri dish. The culture dishes were kept in a water table with running seawater to maintain ambient temperature and were partly shielded to avoid direct light. Water in the culture dishes was changed every 2–3 days and the incubation lasted 5 weeks. Pre-incubation wet weight and arm length (estimated from photographs) were measured for all stars at the start of the experiment, and compared to wet weight, dry weight, body diameter and arm length at the end of the incubation. Each brittle star remaining alive at the end of the experiment was subjected to an agility test where the individual was flipped on its back and the time to return to normal orientation was measured. Wet weight was determined by placing a star on a tissue paper for 2 s before recording its weight ($\pm 0.001$ g). Body diameter and arm lengths were measured to the closest mm, and arm length was averaged over all 5 arms. The brittle stars were dried in a 50 °C oven for 48 h before obtaining dry weight.

### 2.5. Wave Energy Model

In coastal zones and estuaries, temporal and spatial variations in salinity are affected by changes in precipitation, evaporation and freshwater inflows, as well as by water mass mixing related to changes in circulation, waves and tides. Changes in salinity can have major effects on water density and water stratification, which, in turn, can modify circulation patterns. We focused on mixing along open coastlines with water depths similar to those at most discharge sites for proposed SWRO plants in California. We assumed mixing is controlled by wave dynamics and excluded tidal and other effects.

The mixing potential of wave action was quantified by calculating patterns of wave energy and orbital velocity for the mean conditions in each of four seasons, and then averaged them over a year. We used wave model data for the California continental shelf that provides information on wave heights, periods and orbital velocities at high resolution for the whole Californian coast, including mean and extreme (top 5%) wave parameters for each season from [42,43]. A set of 15 SWAN curvilinear grids was used to simulate wind-wave growth and propagation across the inner portion of the California continental shelf [42]. All grids had an average cross- and along-shore resolution of 30 to 50 m and 60 to 100 m, respectively.

An index of annual mixing potential, $E$ was calculated by averaging the mean conditions for each season in Equation (2), where $\rho$ is water density, $g$ the gravitational pull, $H$ is wave height and $L$ wave length (see Supporting Information Equations (S1), (S2) and (S3)):

$$E_L = \frac{1}{8}\rho g H^2 L. \tag{2}$$

We considered seasonal variations in the potential for mixing because the North Pacific Ocean presents extremely large surface waves that can vary greatly between seasons and years [46,47]. Specifically, wave energy varies strongly between winter and summer storms and can also respond to large inter-annual climate variations, for example associated with El Niño events.

In shallow water, the orbital motions of water particles induced by surface waves extend down to the seabed. The resulting wave-induced orbital velocities near the seabed are considered to be representative measures of how waves influence the sea floor [42]. We used the orbital velocity as an indicator of mixing potential at the seabed, based on mean seasonal values from [43] for each season, and then averaged annually, similar to the wave energy ($E_L$).

Using monitoring data available from California Department of Fish and Wildlife [48] coastal ecosystem types were determined (seagrasses, kelp beds), habitats (rocky or sandy substrate), within a 2 km$^2$ radius of the outfall of each existing and proposed seawater desalination plant in Central and Southern California. We mapped the calculated seasonal wave energy together with the ecosystem and habitat type and compared water mixing potential at proposed SWRO sites and the ecological niches that may be affected by SWRO discharge.

*2.6. Statistics and Geospatial Analyses*

All statistical analyses were done using software R (R-studio, version 1.1.447). Salinity, temperature and other water chemical data were pooled in groups depicting the immediate area around the outfall channel (50–200 m offshore) and the surrounding area (250–1000 m offshore) for both surface and bottom water for all four sampling trips. The groups were compared between surface and bottom water with two-way ANOVA and post-hoc tested with Tukey-HSD with a significance level of $\alpha = 0.05$ (Table 1). Biological abundances of epifauna and infauna were similarly pooled and, along with brittle star measurements, analyzed with two-way ANOVAs and post-hoc Tukey-HSD tests ($\alpha = 0.05$). Salinity and temperature were mapped in ArcGIS (ESRI, version 10.2), using Inverse Distance Weighting (IDW) interpolation to visualize temperature and salinity variation. Average annual wave energy was mapped in ArcGIS 10.2 and combined with spatial data on coastal biology and marine habitats [48] including kelp beds, hard substrate (rocky reefs), two seagrasses (eel grass *Zostera*; surf grass, *Phyllospadix*) and marine protected areas (MPA's).

**Table 1.** Mean, maximum and minimum salinity and temperature ($\pm$S.D.), for surface and bottom water in the area immediately around the outfall mouth (50–200 m) and the surrounding area (250–1000 m) measured at the four sampling times. Post-hoc results (Tukey-HSD, $p < 0.05$) for mean salinity and temperature are indicated in small letters for surface water and capital letters for bottom water behind each mean value.

| Sampling Time | Distance from Outfall Mouth (m) | Water Level | Salinity | | | Temperature | | |
|---|---|---|---|---|---|---|---|---|
| | | | Mean | Max | Min | Mean | Max | Min |
| December 2014 | 50–200 (Immediate) | Surface (n = 5) | 33.6 b (0.06) | 33.6 | 33.5 | 18.4 b (0.7) | 19.0 | 17.4 |
| | | Bottom (n = 4) | 33.5 BC (0.01) | 33.6 | 33.5 | 18.2 B (0.8) | 19.3 | 17.1 |
| | 250–1000 (Surrounding) | Surface (n = 5) | 33.4 b (0.1) | 33.5 | 33.3 | 18.8 b (0.6) | 19.5 | 18.4 |
| | | Bottom (n = 4) | 33.5 BC (0.1) | 33.7 | 33.4 | 19.1 B (0.3) | 19.4 | 18.8 |
| September 2015 | 50–200 (Immediate) | Surface (n = 8) | 33.1 b (0.7) | 33.5 | 31.5 | 24.9 a (0.7) | 26.5 | 24.0 |
| | | Bottom (n = 6) | 33.5 C (0.2) | 33.9 | 33.0 | 24.5 A (0.3) | 25.2 | 24.1 |
| | 250–1000 (Surrounding) | Surface (n = 8) | 33.4 b (0.07) | 33.5 | 30.4 | 24.3 a (0.5) | 25.2 | 23.7 |
| | | Bottom (n = 6) | 32.8 C (1.2) | 33.5 | 33.3 | 24.4 A (0.5) | 25.1 | 23.9 |
| May 2016 | Mouth of outfall | Surface | 37.4 a | 37.4 | 37.4 | NA | NA | NA |
| | 50–200 (Immediate) | Surface (n = 7) | 33.9 b (0.7) | 35.5 | 33.6 | 19.1 b (0.4) | 19.4 | 18.2 |
| | | Bottom (n = 6) | 34.9 A (0.7) | 35.7 | 33.9 | 19.3 B (0.8) | 19.8 | 17.8 |
| | 250–1000 (Surrounding) | Surface (n = 7) | 33.4 b (0.5) | 33.8 | 32.5 | 18.8 b (0.6) | 19.4 | 17.9 |
| | | Bottom (n = 7) | 34.4 AB (0.7) | 35.2 | 33.5 | 17.9 B (1.4) | 19.4 | 15.5 |
| November 2016 | 50–200 (Immediate) | Surface (n = 7) | 33.5 b (0.2) | 33.9 | 33.4 | 19.1 b (0.3) | 19.7 | 18.8 |
| | | Bottom (n = 6) | 34.4 AB (0.2) | 35.9 | 34.8 | 19.3 B (0.5) | 20.0 | 18.6 |
| | 250–1000 (Surrounding) | Surface (n = 7) | 33.5 b (0.2) | 33.9 | 33.3 | 19.1 b (0.6) | 19.9 | 18.0 |
| | | Bottom (n = 7) | 34.4 A (0.8) | 35.9 | 33.6 | 18.7 B (0.7) | 19.5 | 17.3 |

## 3. Results and Discussion

### 3.1. Physical and Chemical Characterization of the Discharge Plume

Salinity and temperature (mean, minimum and maximum) in the immediate (50–200 m) and surrounding (250–1000 m) area around the discharge channel of all four sampling times are given in Table 1. Before the Carlsbad Desalination Plant began operations, ambient salinity at Carlsbad Beach was 33.2 $\pm$ 0.6 (pooling all data from December 2014 and September 2015 both surface and bottom water, n = 146). This value is consistent with previous reports of coastal salinity in this region [13]. Surface and bottom salinity for all four sampling times are mapped in Figure 2.

Post-operation measurements show surface water salinity of 37.4 right at the mouth of the outfall channel. In the immediate vicinity (50–200 m) outside the outflow channel, surface salinity averaged 33.9 and 33.5 in May and November 2016, respectively, with a maximum measured surface salinity of 35.5 at ~50 m from the mouth of the discharge channel (Table 1). Surface water salinity did not significantly differ between pre- and post-operation, except for the high salinity water right at the channel mouth ($p < 0.001$) (Table 1).

Bottom water salinity in the immediate vicinity (50–200 m) reached values of 35.9 at 200 m offshore, whereas the average salinity were 34.4 and 34.9 for May and November 2016, respectively, and was significantly higher than salinity in September 2015 ($p < 0.001$) (Table 1).

Further from the channel (250–1000 m offshore), surface salinity was near ambient levels (averaged 33.4 and 33.5 for May and November, respectively) and salinity did not differ significantly between pre- and post-operation (Table 1). The average bottom water salinity in the surrounding area (250–100 m) was 34.2 for both May and November (1 salinity unit higher than ambient), and the maximum bottom salinity of 35.9 was measured at 250 m offshore.

Both surface and bottom water temperatures were 1–2 °C warmer around the outfall than further offshore, or away from the discharge area during December 2014 and May and November 2016 but did not differ between these three sampling times (Table 1). During September 2015, temperatures throughout the coastal area were high due to El Niño conditions with ocean temperatures above normal [49] (Table 1). In May 2016, a thermocline was measured 800–1000 m offshore due to seasonal upwelling (Figure 3G), but the temperature difference between the discharge water and ambient coastal water did not otherwise create thermoclines.

The hydrological mixing model performed for Poseidon Water [13] to estimate the areal extent and dissipation of the plume (footprint) surrounding the outfall incorporated some physical properties of Carlsbad Beach (shallow water and high wave energy), and assumed an initial discharge salinity of 42 at the mouth of the channel. The model predicted sufficient mixing would be achieved to reduce salinity to levels approved for the plant within the 200 m offshore [12,13]. However, our field data indicate that, despite the lower salinity we measured at the outfall compare to that used in the hydrological model (37.4 vs. 42), a discrete salinity plume extends 600 m offshore from the mouth of the outfall, with salinity up to 35.9 observed beyond 200 m offshore (Figure 2C,D,G,H). This is 2.7 units above the mean ambient salinity (33.2) and exceeds the limit of two units over ambient within 200 m of the shore, specified in the California Ocean Plan (2015 Amendment to Water Quality Control Plan). These results raise concerns about the adequacy of the mixing model used, and the higher than expected salinity emphasizes the need for post-operation coastal monitoring. The higher temperature of the plume water (1–2 degrees above ambient) did not lower the density sufficiently to negate the higher salinity and a visible halocline developed close to the discharge channel.

Although the distinct salinity and temperature of the discharge plume were recognizable extending seaward from the discharge channel (Figure 2), the distribution and concentrations of DOC, POM, Chl *a* and nutrients ($NO_3$, $PO_4$ and $SiO_2$) showed no significant temporal or spatial patterns in the region and were not correlated to salinity (see Table S1 in Supporting Information). Concentrations of major elements (normalized to salinity) and trace elements in the plume also did not differ from those in other sampling sites in the vicinity. If our results are representative of SWRO operations at other sites, they suggest that the higher salinity within the plume (perhaps accompanied by higher temperature) would be the dominant chemical factor affecting coastal biota in terms of chemical impacts directly related to the discharge.

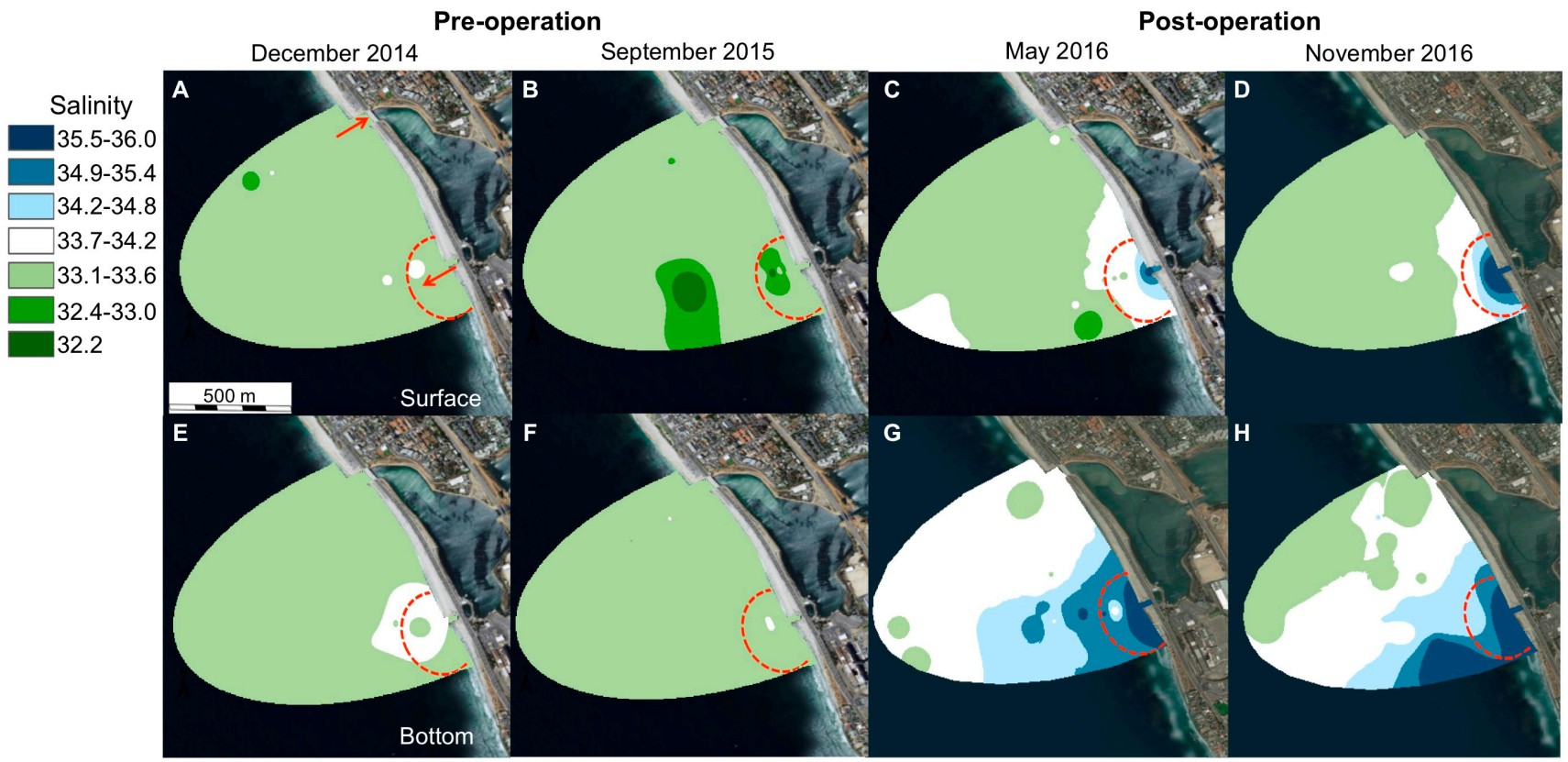

**Figure 2.** GIS-mapped salinity measurements for surface (**upper row**) and bottom water (**lower row**) for pre-operation (**A,B,E,F**) and post-operation (**C,D,G,H**). Arrows in (**A**) indicate the outfall location and lagoon intake area. The red semi-circle represents the 200 m distance set by California Ocean Plan (2015 Amendment to the Water Quality Control Plan).

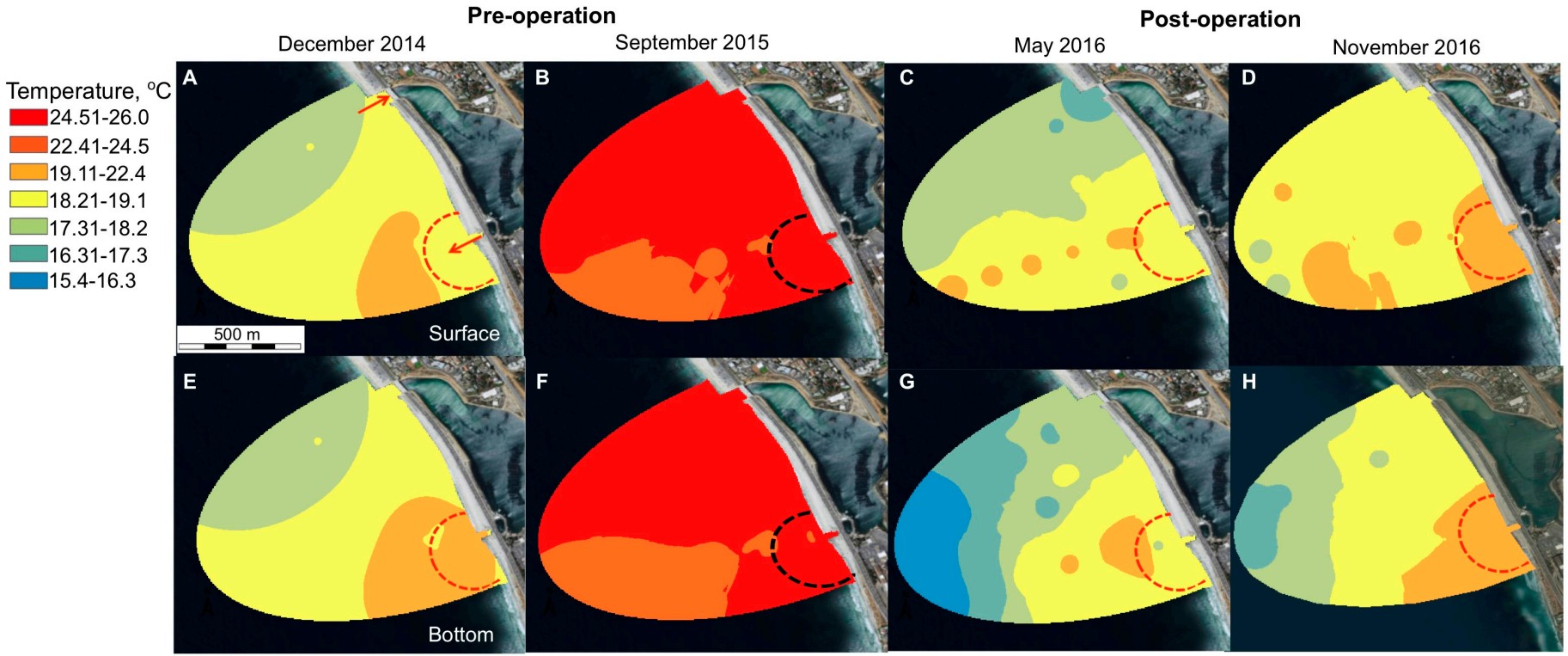

**Figure 3.** GIS-mapped temperature measurements for surface (**upper row**) and bottom water (**lower row**) for pre-operation (**A**,**B**,**E**,**F**) and post-operation (**C**,**D**,**G**,**H**). Arrows in (**A**) indicate locations of the outfall and lagoon intake. September 2015 was an El Niño year with higher average temperatures in the coastal zone (**B**,**F**). The red semi-circle (black for Sep 2015) represents the 200 m distance set by the California Ocean Plan (2015 Amendment to Water Quality Control Plan).

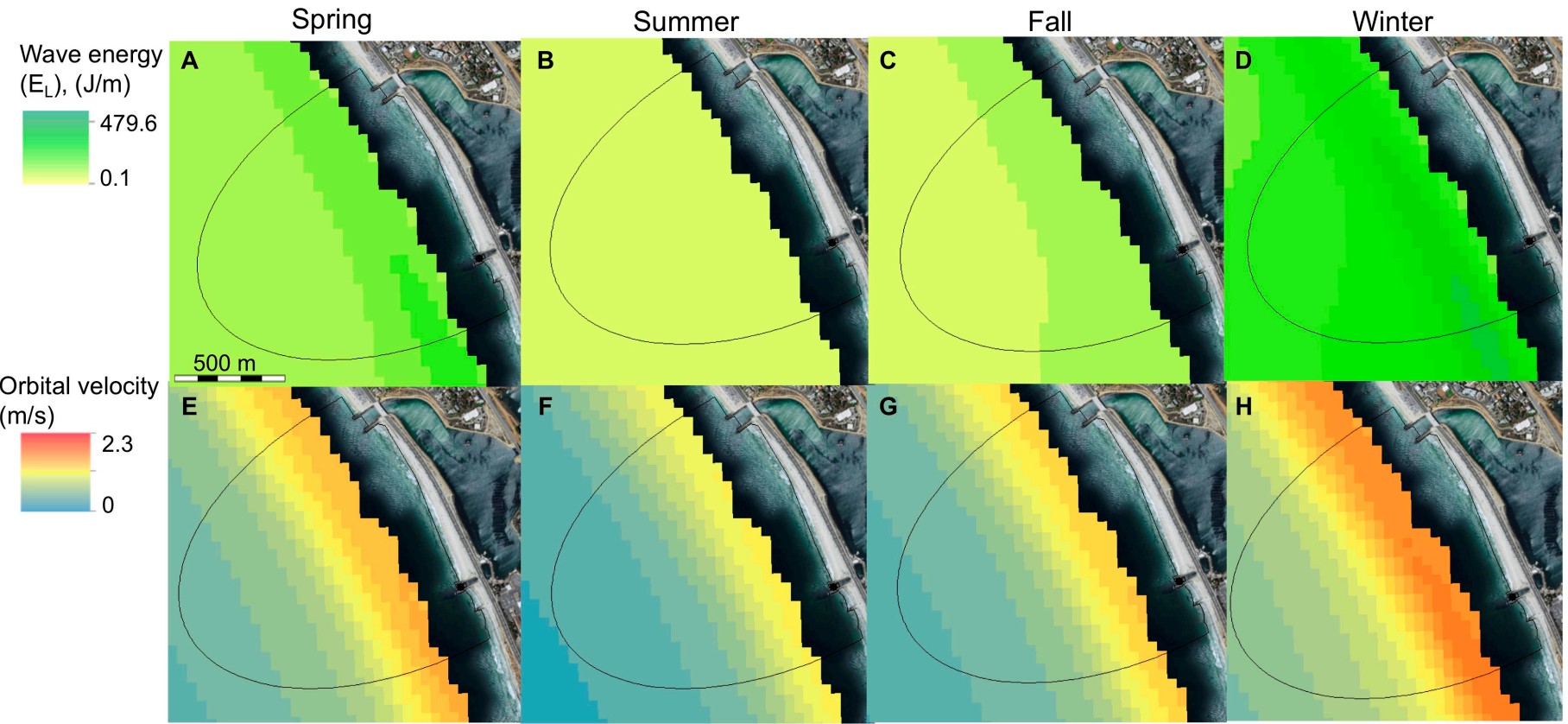

**Figure 4.** GIS-mapped averaged seasonal wave energy density (upper row **A–D**) and orbital velocity (lower row **E–H**) during spring, summer, fall and winter. The black parabola indicates the extent of the area of our field samples.

While the density of discharged brine and brine dispersion can be controlled by the design and operational practices of SWRO plants, coastal bathymetry and wave energy that also affect plume dispersal are not controlled through plant design. However, this study shows that these parameters should be considered when selecting discharge locations. Carlsbad Beach is a high energy site (Figure 4) where conditions appear optimal for brine dilution via extensive mixing in the nearshore area [13,40,41], yet the plume was clearly detected at least 600 m offshore. Hence, dilution via mixing at this site is inadequate. Wave energy and orbital velocities vary seasonally (Figure 4); therefore, wave-driven mixing potential should also vary throughout the year, with greater mixing in winter—however, we detected little difference in the brine plume's extent and properties between May and November 2016. This suggests that these seasonal mixing differences were not sufficient to alter the spatial extent of the brine plume. Further dilution of the brine prior to discharge (currently 1:10 with cooling seawater) would be required to lower the plume density and increase mixing potential. For future plants, this could be achieved if greater volumes of water are used for cooling in a co-located power plant or if combined with other freshwater discharges like sewage discharge. As physical and oceanographic properties are not adjustable, use of diffusor systems instead of open channels offers a solution for increasing dilution and mixing of SWRO discharges. At several SWRO facilities worldwide, implementation of diffusors has successfully increased brine mixing [50,51]. Continuous monitoring over several months or years would be necessary to determine whether the Carlsbad salinity plume always extends far from shore, or if it was due to special conditions at the time of our sampling.

*3.2. Benthic Macrofauna and Ecotoxicological Study of Brine Exposure*

The benthic epifauna had higher proportions of species at the northern end of the beach than at the southern end and around the discharge channel (see Supporting Information Figure S1), mainly due to different habitat types (small rocky reef in the north versus sandy bottom elsewhere). Accordingly, we only compared abundances at sampling sites with similar substrate pre- and post-operation.

The epifauna around the discharge area was dominated by tube-forming polychaetes (Onuphidae). Post-operation the abundance of Onuphidae was significantly higher in the immediate vicinity of the discharge channel (i.e., <200 m from the channel mouth) than in sandy areas away from the outfall ($p < 0.001$) (Figure 5A). The abundance of epifauna was significantly higher ($p < 0.001$) in the immediate vicinity of the discharge channel (0–200 m) compared to the abundance in the surrounding area (250–1000 m) in post-operation measurements (7.33 ± 3.8, 4.6 ± 0.9 and 1.4 ± 0.6, 1.6 ± 0.7 individuals m$^{-2}$ for May and November 2016 in close vicinity and surrounding area, respectively (median ± SD)) (Figure 5A). This contrasts with pre-operation measurements, in which epifauna were significantly more abundant in the surrounding sandy areas (250–1000 m from discharge) than close to the channel (0.5 ± 1.3 individuals m$^{-2}$ in the 0–200 m zone for both December and September and 7.3 ± 2.9 and 28 ± 15.7 for the 250–1000 m, December and September, respectively).

Infaunal organisms (in the upper ~10 cm of sediment) were generally more abundant in post-operation samples, but the difference was not significant (Figure 5B). The BOPA index close to the outfall channel was high (average of 0.23 ± 0.07), indicative of a poor ecological area with high disturbance. However, BOPA values did not differ between pre- and post-operation sampling times (Figure 5C).

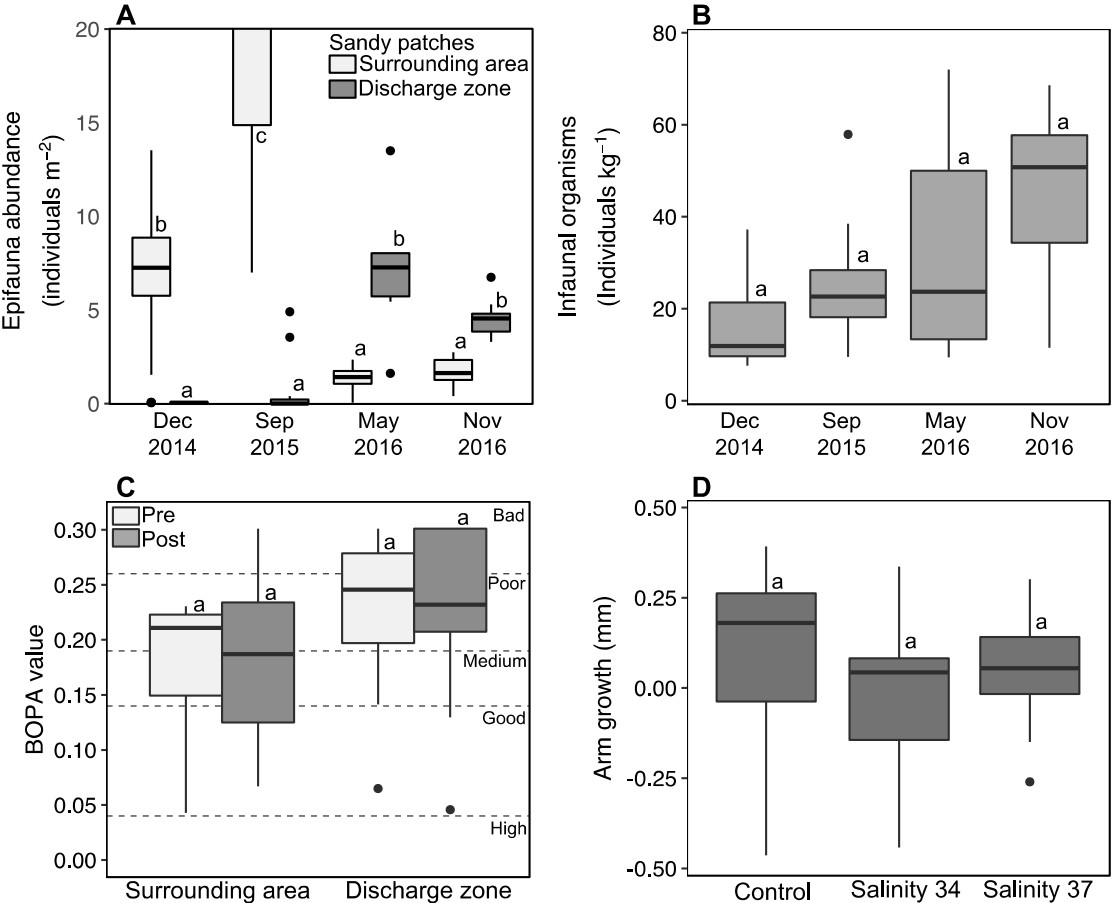

**Figure 5.** Box-whisker plots with significance letters above boxes refering to ANOVA followed by a Tukey-HSD test with significance of *p* < 0.05. (**A**) Median values for epifauna abundance in the immediate discharge zone (50–200 m from shore) and in the surrounding sandy areas (250–1000 m offshore), and (**B**) infaunal abundance in the area around the outfall channel (100–1000 m offshore) for pre-operation (December 2014 and September 2015) and post-operation (May and November 2016). Box-whiskers plot of (**C**) BOPA-values of pre- and post-operation at the sandy areas in the discharge zone (50–200 m from shore around the outfall) and in the surrounding sandy areas. The horizontal dashed lines indicating the ecological rank from good (high biodiversity) to bad (low biodiversity for mainly opportunistic species). Box-whiskers plot of the brittle stars' arm growth (**D**) after five weeks of incubation in brine from the outfall of Carlsbad Desalination Plant.

Grain size analysis of the sediment surrounding the outfall gave a sorting value of σ = 0.8 ± 0.2 both pre- and post-operation, indicating moderately sorted sand implying high flow rate from the outfall channel which was indeed easy to observe from shore [45] (see Supporting Information Figure S2). Grain size away from the outfall has a larger fraction of fine grains being less impacted by channel flow. It is likely that the focused discharge of cooling water from the power plant for the past ~65 years has resulted in sand winnowing and extensive local disturbance of the sediment and benthos, and the addition of brine from the desalination plant does not induce any measurable acute toxicity for either epifaunal or infaunal organisms within the plume. In fact, the increase in polychaetes abundance at the outfall channel may be due to differential survival of organisms in the salinity plume, perhaps due to reduction of predation by larger organisms that may avoid the salinity plume [31,52]. While the lower abundance of epifauna in sandy areas away from the outfall zone supports this idea, we cannot rule out possible effects of natural seasonal variability or differential species recruitment. Particularly, the occurrence of El Niño during 2015 could have been a main driver for changes in abundances and recruitment.

Regardless of the specific causes, the already disturbed area from the cooling water discharge from the power plant, and the generally low biodiversity of organisms in the sandy environment (see Supporting Information Figure S1), result in minimal direct impacts of Carlsbad Desalination Plant on the ecology of benthic organisms still inhabiting this area, despite the salinity plume extending >200 m offshore.

In the incubation experiment, the brittle star growth (arm length) was not significantly different between the treatments (Figure 5D, $p > 0.06$). There also was no impact on mortality. Brittle stars are robust organisms present in virtually every marine habitat [53], and our data suggest that adult brittle stars can live at higher salinity than ambient, including salinity expected very close to discharge plumes [53]. In sandy environments like Carlsbad Beach, brittle stars are important for causing bioturbation in the upper sediment that enables rapid nutrient and oxygen flows to the sediment [54,55]. The growth and development of other echinoderms endemic to coastal California (*Strongylocentrotus purpuratus* (Purple Urchins) and *Dendraster excentricus* (Sand Dollar)) also are not significantly impacted by salinity increases up to 37 (~10% above ambient salinity of ~33) [36].

Observations around other desalination facilities worldwide have linked decreased abundances of epi- and infaunal organisms to increased salinity, typically when it reaches 5% above ambient [17,23,56]. Our field observations have not detected measurable negative changes (e.g., decrease) when comparing the pre- and post-operation conditions. As noted above, we speculate that this is likely because the site was already impacted by discharge from the power plant. We have not investigated organisms that are present in other settings in coastal California (for example, kelp or sea grasses) and it is possible that other organisms may be sensitive to the 3–5% above ambient salinity seen within the plume at Carlsbad Beach.

### 3.3. Future Desalination Plants in California

Rocky reefs, kelp beds and seagrass beds are widespread along the coast of California, and these highly productive ecosystems can be sensitive to small changes in their environments [57,58]. Previous experiments have shown higher mortality and lower sporulation in the macroalga *Ulva pertusa* when salinity was increased by 5–10% over ambient [25]. Similarly, seagrasses are particularly sensitive to salinity increases [20,23,59]. Our data indicate that SWRO discharge has little impact on organisms inhabiting sandy seafloors, particularly in non-pristine anthropogenic impacted areas, and such sites should be preferable locations for discharge of brine from new SWRO facilities. In addition, selecting sites with high wave action will increase mixing and dilution of brine discharges; discharge should be made offshore and in areas subjected to wave action that favors mixing.

At Carlsbad Beach, despite the relatively high average wave energy conditions (Figure 4 and Supporting Information Table S2), wave-driven mixing was not sufficient to reduce salinity to less than two units above ambient within 200 m from the discharge point. This suggests that meeting California's requirement for maintaining salinity of no more than two units above ambient within 100 m from the discharge site [12] at new SWRO facilities, particularly large ones (e.g., Camp Pendleton), will need more efficient discharge methods such as offshore diffusor systems, rather than a point discharge near the beach. Our modeling of proposed sites in Southern California (Camp Pendleton and Dana Point) shows they have lower wave energy than Carlsbad, and hence wave mixing is likely to be less efficient close to shore (Figure 6). At Camp Pendleton, the coastal substrate is dominated by sand, which will likely minimize possible ecological effects of the brine discharge; however, the coastal area surrounding Dana Point is dominated by rocky reefs that may hold great biodiversity, and caution should be taken when discharging brine at this site.

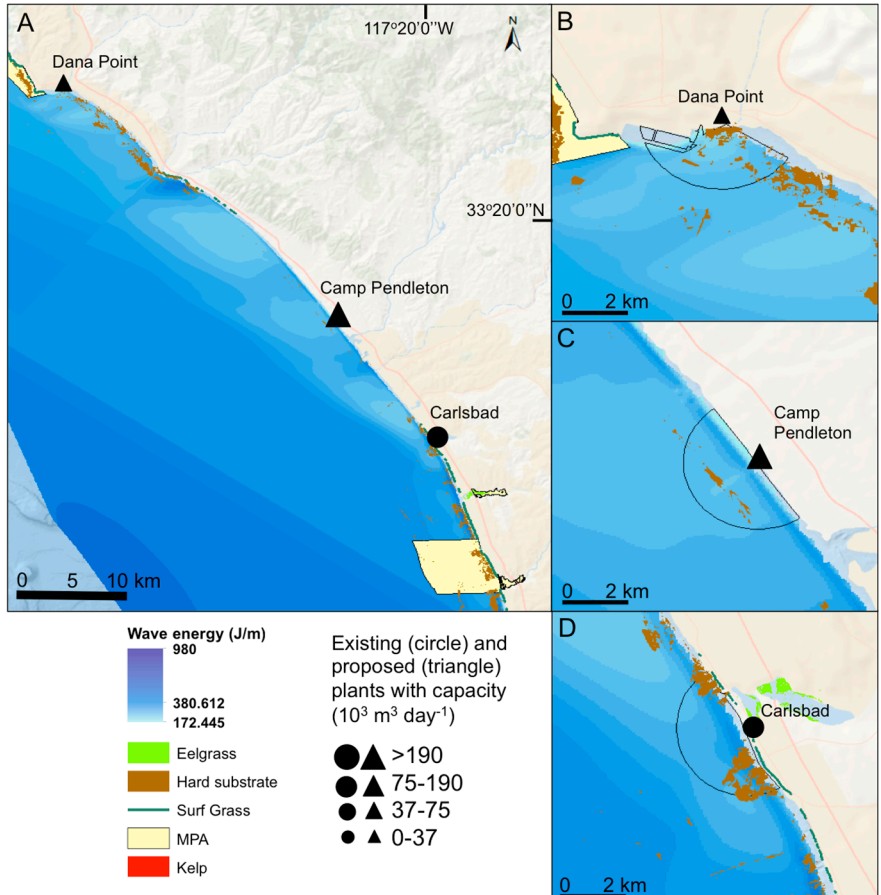

**Figure 6.** Average wave energy and benthic habitat for southern California with existing (circles) and proposed (triangles) desalination facilities (**A**). Zoomed perspective is shown in **B–D**.

All proposed sites in Central California, except for the Monterey Bay Peninsula Plant, are lower wave energy environments than at Carlsbad Beach (Figure 7 and Table S2) and presumably, lower mixing potentials. All of these plants are designed to have lower capacities than the Carlsbad Plant and, for the proposed plant at Moss Landing, co-location with an existing power plant and combining brine discharge with cooling water to increase dilution prior to discharge is possible. The benthic habitat or the three proposed locations around Monterey Bay is a predominately sandy bottom, but the two sites close to Moss Landing are near Elkhorn Slough, a protected National Estuarine Research Reserve, where a more cautious approach with extended monitoring of the brine mixing may be needed (Figure 7).

It is paramount that all proposed future desalination plants have comprehensive modeling of mixing trends that include seasonal differences in tides, waves and wind conditions at those sites and study the mixing potential in detail to assess if other techniques for diffusion are needed. To reduce the footprint of brine plumes and their possible impacts, use of alternative discharge technologies, such as diffuser systems, should be considered and appropriate modeling of the discharge design used. Both models and practice indicate that diffuser systems increase the mixing of discharge brine with ambient seawater [50,56,60,61]. Our results also suggest that long-term monitoring of coastal areas in the vicinity of proposed plants will be needed both before and after construction and operation. Ecosystem state and the likelihood for negative impact has to be considered in decisions regarding the location of future plants with effort to ensure that discharge will occur in areas with already low biodiversity (e.g., sandy areas and not near rocky substrate, kelp or seagrass) and where previous disturbances have already resulted in selection for resilient organisms (e.g., co-locating with power plant discharge, dredged areas or other potential natural or man-made disturbances).

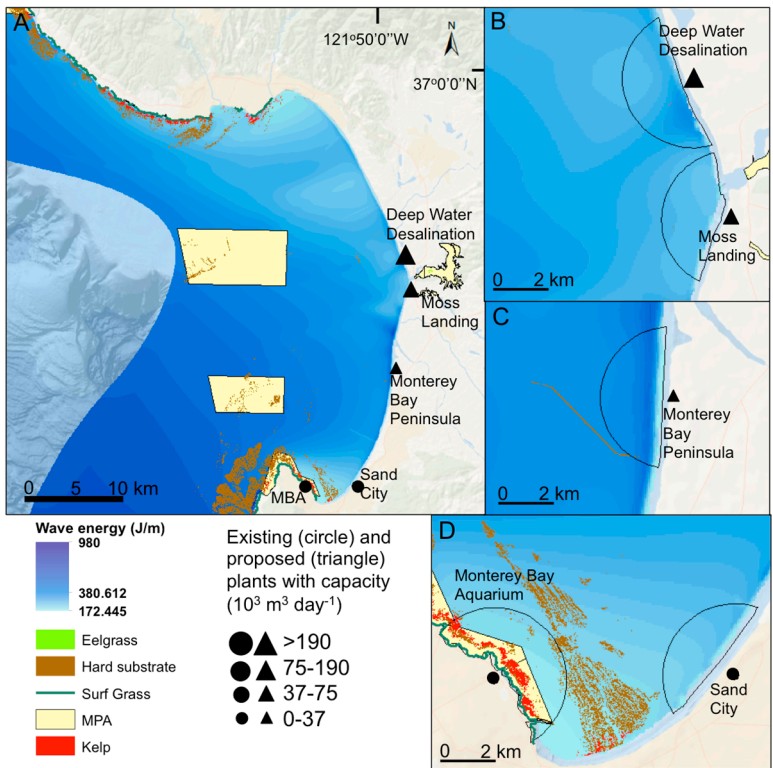

**Figure 7.** Average wave energy and benthic habitat for Monterey Bay with existing (circles) and proposed (triangles) desalination facilities (**A**). Zoomed perspective is shown in **B**–**D**.

## 4. Conclusions

A distinct salinity plume was observed around the outfall of Carlsbad Desalination Plant with a salinity increase of up to 2.7 units above ambient and extending 600 m offshore. This exceeds the maximum allowed salinity perturbation in the coastal zone as stated by the California Ocean Plan (2015 Amendment to Water Quality Control Plan). However, no significant impact was found on the benthic epifauna or infaunal composition and abundance in the same area, possibly due to an already disturbed conditions in this setting due to previous and ongoing discharge of post-cooling water from a local power plant. The coastal wave energy and mixing potential at Carlsbad Beach is high compared to other locations along the Southern and Central California coastline, and the lack of sufficient mixing of the brine at Carlsbad Beach raises concern regarding the efficiency of brine mixing at other proposed sites for desalination plants along the coast of California.

We recommend all future desalination facilities to be designed with optimal discharge options of either diffusor systems or a higher dilution of the brine prior to discharge. Furthermore, they should, where possible, be co-located with other water discharging facilities such as power plant or waste water treatment plants to restrict potential impacts to areas that are already disturbed from anthropogenic activities. Finally, we stress the need for long-term monitoring of coastal ecosystems in the area surrounding proposed and current desalination facilities to fully understand the ecological response to long-term exposure to salinity increase.

**Supplementary Materials:** The following are available online at http://www.mdpi.com/2073-4441/11/2/208/s1, Figure S1: Benthic macrofauna. Figure S2: Sediment grain size distribution, Table S1: Water chemical properties. Table S2: Brittle star growth. Table S3: Wave energy and benthic habitat estimation.

**Author Contributions:** Conceptualization, K.L.P., D.P. and A.P.; Data Curation, K.L.P. and A.P.; Formal Analysis, K.L.P., N.H., B.G.R. and A.H.; Funding Acquisition, D.P. and A.P. and K.L.P.; Investigation, K.L.P., D.P. and A.P.; Methodology, N.H. and B.G.R.; Project Administration, A.P. and K.L.P.; Resources, A.P.; Software, N.H. and B.G.R.; Supervision, D.P. and A.P.; Validation, D.P.; Visualization, K.L.P., N.H. and B.G.R.; Writing—Original Draft, K.L.P.; Writing—Review and Editing, K.L.P., N.H., B.G.R., D.P., A.H. and A.P.

**Funding:** This work was supported by National Science Foundation COASTAL SEES #1325649 awarded A.P and D.P. K.L.P. was partially supported by student fellowships from the Geological Society of America (#: 11215-16), the Weigel Scholarship for Coastal Studies, Myers Oceanographic and Marine Trust and the Explorers Club Youth Foundation (#: 83402).

**Acknowledgments:** We would further like to acknowledge the students and researchers of the Paytan Lab at UCSC, in particular Joseph Murray, Ana Martinez Fernandez, Kim Bitterwolf, Kyle Broach and Katie Roberts and the scientific divers and boat crew, especially Jacque Lord and Rich Walsh, for their extensive help and support in completing this work.

**Conflicts of Interest:** The authors declare no conflict of interest.

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
