# Peer review of "Biological and Physical Effects of Brine Discharge from the Carlsbad Desalination Plant and Implications for Future Desalination Plant Constructions"

_water, doi:10.3390/w11020208_

Round 1

Reviewer 1 Report

In this article, the author investigated the effects of continuous discharge of high-salinity brine into coastal environments near Carlsbad Desalination Plant in California. The author found that the spatial footprint of the brine discharge plume extended about 600 m offshore and the bottom salinity exceeds the maximum salinity permitted for this location based on the California Ocean Act. However, the author claims there was no significant changes in the assessed biological indicators. The author also gave suggestions to the potential desalination plants on how to choose locations and how to treat the brine in CA.

The results from this case study are useful for central and southern CA. The only concern I am having is the accuracy/representative of the data. From section 2.2, the author only took 1 water sample (1L) and 1 sediment sample (250ml) at each sampling site. And the visible benthic epifauna were enumerated separately by two scuba divers. It seems to me that the data was easily affected by the weather, wave and other conditions.

Also, Please add unit for salinity (ppt). and the two equations (1 and 2) should have the same format.

Author Response

See response in attached pdf file.

Reviewer 2 Report

This study “Biological and physical effects of brine discharge from Carlsbad Desalination Plant and implications to future desalination plant constructions” is a fairly robust and well-organized study that warrants publication. I especially appreciated the multi-year effort that allowed for a before-after comparison.  I also understand the importance of this study in terms of tracking the impact of industry on the environment. There are a few minor issues and corrections that should be addressed before this study is finalized and published. See the following below:

53: spelling error- discharge

74: Consider putting global impacts in first paragraph rather than between info related to California.

104: Consider revising sentence by splitting it up in two parts.

107: spelling error – illustrate

107: Consider not using “We”

107: The remainder of the introduction sounds too much like a methods summary. Although most authors may find this odd, this journal wants you to highlight conclusions here. From the MDPI website instruction to authors - “Introduction: The introduction should briefly place the study in a broad context and highlight why it is important. It should define the purpose of the work and its significance, including specific hypotheses being tested. The current state of the research field should be reviewed carefully and key publications cited. Please highlight controversial and diverging hypotheses when necessary. Finally, briefly mention the main aim of the work and highlight the main conclusions. Keep the introduction comprehensible to scientists working outside the topic of the paper.”

120: Use of comma reads oddly – “The Carlsbad Desalination Plant, built and operated by Poseidon Water, is the first and, currently, the only large-scale SWRO desalination facility in California.”

126: “post cooling” or post-cooling?

140: Why did you decided to not bracket the outfall with perpendicular transects on both sides? It seems that the actual extent was not captured in this sampling design. Was this a current direction issue? If so, please describe this current and how it impacted sampling design.

155: What type of filters were used?

160: Be consistent with placement of units after a value “1000m”.

161: Describe what type data was produced. Did you count all organisms of a particular category or was it merely presence? Units?

164: Describe how epifauna were enumerated. Units?

173 and elsewhere: Pore size of filters?

179: Be consistent: NO3 and SiO2 or nitrate etc.

211: Why is (Monterey Bay Abalone Co) shown as a reference for the Bay?

215: How was dissolved oxygen controlled? Did you use a bubbler to add air or monitor DO?

257: How did you determine ecosystem types? Observation? Previous studies? Was this in the methods?

313: spelling – compare

336: recommend not using at least. Give the furthest distance a definite impact was found.

353: The legend text appears cut for the top row red color when viewed at 100% but disappears as I zoom in on the page.

356: Maybe choose another color for the 200 m distance set by the CA Ocean Act. It is hard to see in contrast to Sept. 2015 data.

357- Why is scale for Wave energy so large when the does not seem to be any blue in the images? Consider rescaling for better resolution. Units for orbital velocity?

363: Consider breaking into two sentences.

403: Reference 53 seems out of place and duplicate.

430: The common name Giant Kelp is typically a reference to Macrocystis pyrifera, a brown seaweed. Ulva pertusa is a smallish green seaweed.

Author Response

See response in attached pdf file

Reviewer 3 Report

The manuscript by Lykkebo Petersen et al. on the effects of desalination brine discharge at a plant in Carlsbad, CA, and their implications for other plant operations was very well written, scientifically solid, advances our knowledge in this field, and was a pleasure to read.

I, therefore, recommend its publication with minor edits, that I describe below.

Before I do so, I'd like to mention that if this work is to be continued, more attention should be placed on pooling of brine within the seafloor and sedimentary biogeochemistry, especially since extensive sand fields can be found at this site. I agree that sandy bottoms are likely to minimize the impact of brine on surrounding communities, to a large degree because of their permeability, hence the maximization of the volume over which discharges can mix with ambient water. I also do understand that monitored properties were selected with a plethora of considerations, and I don't mean this comment to impact anyone's view of this great study. It's just something to think about if a follow-up is in the works.

I also have a general Policy-related point. Throughout the manuscript, there is mention of the "California Ocean Act," while in line 64 there is mention of the "California Coastal Act (2015)". I am aware of the:

Desalination Amendment of 2015 to the Water Quality Control Plan for the Ocean Waters of California, often referred to as the Ocean Plan

California Ocean Protection Act of 2004

California Coastal Act of 1976

Could the authors please modify their mentions (all instances) in their manuscript so that it is clear which provisions they are referring to?

As for the minor edits, for ease of navigation, I will refer to specific pages and lines.

In the abstract, in lines 20-21, it is mentioned that "We collected in situ measurements ... in coastal waters (~2 Km from shore)." After having read the paper, I think that the text in the parentheses would more accurately read "(up to ~ 2 km from shore)".

In lines 155 and 157, there is mention of "(section 0)". Could you be referring to a section of the manuscript? Please correct or omit.

Mention of the "already disturbed area" in line 395 can baffle someone like me who only got it after getting two paragraphs down, remembering that a large power plant has been discarding cooling water there. Modify the phrase as follows or similarly for clarity: "already disturbed area from the cooling water discharge of the power plant".

Please qualify the statement "the generally low biodiversity of organisms in the sandy environment" in lines 395-6 with a citation, if you are referring to sand as a habitat in general, or a clarification, if you are referring to your own data.

Is further dilution, mentioned in line 341-342 feasible? Engineering-wise it seems to be difficult. A brief comment here on this would be very enlightening.

In the last paragraph on page S4 of the supplement, there is mention of an inset ("see inset"). I'm not sure what this is in reference to.

Author Response

See response in attached pdf file
